



# New Controls on Sedimentation and Climate in the Central Equatorial Pacific Ocean

Allison W. Jacobel[1], Kassandra M. Costa[2], Lily M. Applebaum[1], and Serena Conde[2,3]

1 Department of Earth and Climate Sciences, Middlebury College, Middlebury, 05734, USA
2 Department of Geology and Geophysics, Woods Hole Oceanographic Institution, Woods Hole, 02543, USA
3 Department of Earth, Atmospheric and Planetary Sciences, Massachusetts Institute of Technology, Cambridge, 02139, USA
*Correspondence to*: A.W. Jacobel <ajacobel@middlebury.edu>

**Abstract.** The equatorial Pacific is a nexus of key oceanic and atmospheric phenomena, and its regional climate has critical implications for hydroclimate, the partitioning of $CO_2$, and temperature on a global scale. The spatial complexity of climate signals across the basin has long posed a challenge for interpreting the interplay of different climate phenomena including changes in the Intertropical Convergence Zone (ITCZ) and El Niño Southern Oscillation. Here, we present new, millennially resolved sediment core chronologies and stable isotope records from three sites in the equatorial Pacific's Line Islands region, as well as updated chronologies for four previously studied cores. Age constraints are derived from $^{14}C$ (n=17) and $\delta^{18}O$ (n=610), which are used as inputs to a Bayesian software package (BIGMACS) that constructs age models and uncertainty bounds via correlation with the global benthic $\delta^{18}O$ stack (Lee et al., 2023). We also make use of the new planktonic $\delta^{18}O$ data to draw inferences about surface water salinity and to infer a southward-shifted position for the ITCZ at the Last Glacial Maximum (18-24 ka) and Marine Isotope Stage 6 (138-144 ka). These new chronologies and related datasets improve our understanding of equatorial Pacific climate and show strong promise for further surface and deep ocean paleoclimate reconstructions over the last several glacial cycles.

## 1 Introduction

The equatorial Pacific Ocean is a region of critical importance for the global climate system. Excess heat and moisture from the region are crucial drivers of global atmospheric convection (De Deckker, 2016) and biogeochemical and oceanographic conditions in the region are a sensitive determinant of ocean/atmosphere carbon partitioning (e.g., De Deckker, 2016; Feely et al., 2004; Martínez-Botí et al., 2015; Takahashi et al., 2002). The equatorial Pacific has also been shown to induce significant high-latitude variability in ice sheets via teleconnections (e.g., Clem et al., 2019; Criscitiello et al., 2014; Ding et al., 2011; Steig et al., 2013; Trenberth et al., 1998).

Despite the importance of the equatorial Pacific for understanding global climate, a significant obstacle to its characterization has been that internal dynamics result in spatially complex responses that limit the extrapolation of reconstructions, especially from margin-proximal sites in the east and the west where the data density is higher. For example, characterizing changes in the frequency of extreme El Niño-Southern Oscillation (ENSO) events under different climate states is valuable because of the phenomenon's control of near-field (Trenberth et al., 1998) and distal (e.g. Dai and Wigley, 2000) climate, but the spatial





complexity of the ENSO phenomenon requires data from more than one site to effectively characterize change. Recent work, synthesizing both paleoclimate data and model results, illustrated that the central equatorial Pacific (CEP), including the Line Islands, is a critical region in which to detect changes in ENSO variance (Thirumalai et al., 2024). The importance of providing a spatial perspective on past changes in ENSO patterns is also exemplified by observational work that has identified 'flavors' of ENSO, specifically ENSO Modoki which has a unique fingerprint in the CEP (Ashok et al., 2007). New sediment core

chronologies from this understudied region provide an important foundation upon which to build future high-resolution work, including on climate states warmer than present.

The Central Equatorial Pacific (CEP) is an ideal location to study the coupled ocean-atmosphere climate phenomena of the Intertropical Convergence Zone (ITCZ) (e.g., Jacobel et al., 2017a, 2016; Reimi et al., 2019; Reimi and Marcantonio, 2016).

Precipitation associated with the ITCZ's atmospheric convection varies seasonally and on longer timescales in response to the hemispheric thermal balance (Donohoe et al., 2013; Schneider et al., 2014) with implications for tropical hydroclimate (e.g. Yancheva et al., 2007), global atmospheric circulation (Ceppi et al., 2013), and associated oceanographic upwelling and carbon release (Anderson et al., 2009; Denton et al., 2010). In the CEP the seasonal range of the modern ITCZ is relatively narrow, in contrast with the margins of the basin, where the boundaries of the ITCZ are strongly influenced by topography and seasonal

changes in the land-ocean heat contrast. At the same time, observations suggest that the CEP is one of the areas most sensitive to changes in mean ITCZ position, experiencing a regional ~5° latitude shift in mean ITCZ position when the global average shift is just 1° (McGee et al., 2014). These observations suggest that the CEP is unusually sensitive to small changes in the ITCZ, making global shifts easier to detect, even if their magnitude is not directly interpretable as representative of the global mean.


High accumulation rate sites that are continuous over the past few glacial cycles are the most useful for providing insight into the variability of ENSO mean state and mean ITCZ position. Considering bioturbation depth scales on the order of 6-10 cm (Boudreau, 1994), accumulation rates on the order of 3 cm/kyr or more are necessary for high temporal resolution. At lower accumulation rates, or higher bioturbation depth scales, sedimentological and climatological signals can easily be smoothed,

limiting our ability to understand the timing of events, leads and lags, and to reconstruct the true amplitude of proxy signals and thus associated climate phenomena. In 2012, a geophysical surveying and coring expedition aboard the R/V Marcus G. Langseth, cruise ML1208 (summarized in Lynch-Stieglitz et al., 2015), took 39 cores extending along the Line Islands ridge from just south of the equator to ~8°N and along the NW to SE trending line of atolls spanning 162°W to 154°W. Seismic surveying was used to identify high-accumulation rate, stratigraphically continuous sedimentary sequences from a range of

depths (Lyle et al., 2016). Preliminary age model work on these cores identified sedimentation rates in the range of 1-4 cm/kyr (Lynch-Stieglitz et al., 2015) with significant local variations in sedimentation as a function of dynamic reworking of sediments via bottom currents (Lyle et al., 2016).





Here, we present new stratigraphic controls for three ML1208 cores 06BB (6.24 N, 161.45 W; 2,371m), 16BB (0.48 N, 156.27 W; 2,926m), and 35BB (6.40 N, 160.44 W; 3,777m), and new percent coarse fraction (%CF) data from 26MC/25BB (2.46°N, 159.39W, 3,545m) and 29MC/28BB (2.97°N,1 59.20W, 3,152m) (Fig. 1). These cores were selected to sample a range of water mass depths and surface productivity conditions (basemap, Fig. 1), to capture changes in carbonate preservation as a function of glacial changes to water mass characteristics, and to provide companion cores for previously studied ML1208 cores that have become depleted in key intervals. We use stable oxygen isotope data from planktonic foraminifera to constrain stratigraphy, and we complement these data with radiocarbon dates to generate well-constrained age models with statistically robust estimates of uncertainty.

This study also reevaluates data from seven cores for which radiocarbon- and/or oxygen isotope-based age constraints have previously been developed (Costa et al., 2016; Jacobel et al., 2017a, 2016; Lynch-Stieglitz et al., 2015; Monteagudo et al., 2021) providing new age models for ML1208-15GC (0.16° N, 156.12° W, 3,597m), 18GC (0.6 N, 156.7 W, 3,362m), 20BB (1.3 N, 157.3 W, 2,850m) and 28BB (3.0 N, 159.2 W, 3,153m), and new synthesis interpretations that include existing data from 17PC (.5 N, 156.5 W, 2,926m), 31BB (4.7 N, 160.1 W, 2,857m) and 37BB (7.0 N, 161.6 W, 2,798m). Specifically, we use planktonic foraminifera δ$^{18}$O in tandem with previously published Mg/Ca-derived sea

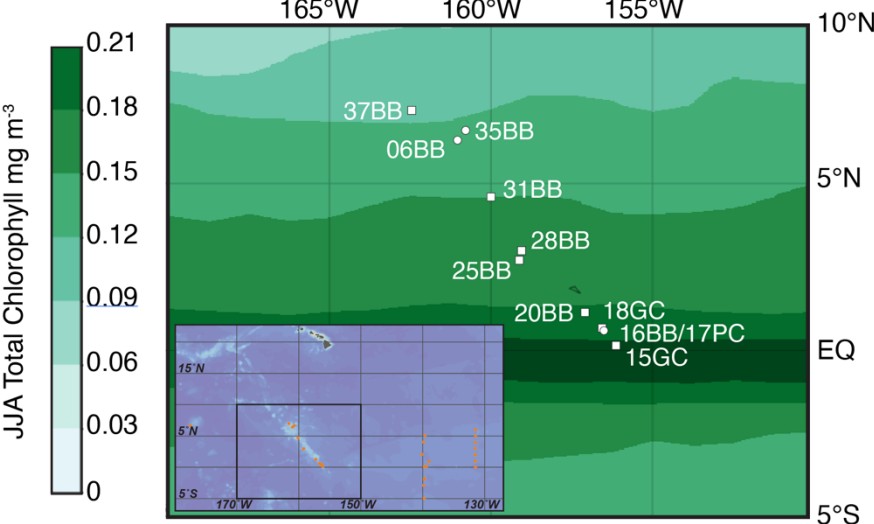

**Figure 1. Map of the Line Islands study sites that are the focus of this work.** Dots show sites with new δ$^{18}$O and $^{14}$C data and squares indicate sites with previously published age constraints where we present new %CF data and/or updated age models. Basemap shows satellite-derived chlorophyll concentrations for a representative June, July, and August (JJA). Data are from Rousseaux & Gregg (2014) plotted using NASA's Giovanni (Acker & Leptoukh, 2007). Inset shows the location of the Line Islands sites (box) in the broader CEP context with a bathymetric basemap. Of the core sites pictured, only records from the Line Islands and those from 72PC extend beyond the last glacial maximum. Inset map created using Ocean Data View (Schlitzer, 2016).

surface temperature to draw conclusions about the movement of the ITCZ over the last two glacial cycles. These conclusions make the dual use of standard 'stratigraphic data' for chronologic and climate reconstruction related purposes.

## 2 Regional Setting and Previous Work

The Line Islands are located south of Hawaii, and just north of the equator (Fig. 1). The islands trend northwest to southeast and are an ideal location for sediment coring, in part, for their height above the regional carbonate compensation depth of



~4,800 m (Lyle et al., 2016) (Fig. 1, inset). The bathymetric range of the ML1208 cores means that it is possible to retrieve carbonate-rich sediments from the ridge flanks, spanning a variety of depths that intersect with abyssal Pacific water masses including Pacific Deep Water (PDW) at intermediate depths (1,600 -3,800 m) and Lower Circumpolar Deep Water (LCDW) at the deepest points (3,800 m). Sediments from the ridge thus have significant potential to capture past changes in deep watermass characteristics and geometry (e.g., Jacobel et al., 2020, 2017a; Lynch-Stieglitz et al., 2015), in addition to surface ocean-atmosphere hydroclimate phenomena (Costa et al., 2017, 2016; Jacobel et al., 2017a, 2016; Monteagudo et al., 2021; Reimi et al., 2019; Reimi and Marcantonio, 2016; Rongstad et al., 2019; Rustic et al., 2015; White et al., 2018). In contrast with other CEP core sites (Fig. 1, inset) where reconstructions only extend to the last glacial maximum, records from the Line Islands have been shown to cover the last several glacial cycles (Jacobel et al., 2017a; Lynch-Stieglitz et al., 2015) and here we expand the available temporal perspective. The other well-studied records from the CEP that cover multiple glacial cycles are from cruise TTN013 (~140˚W) (e.g., Anderson et al., 2019, 2008; Winckler et al., 2008), but the water depths of these cores are significantly deeper than the Line Islands and have relatively low sedimentation rates on the order of 1.5 cm/kyr (Anderson et al., 2008).

The latitudinal range of the Line Islands encompasses biogeochemical variability due to equatorial upwelling and surface current systems. Although the precise latitudinal bounds of current systems fluctuate as a function of seasonal and interannual variability (e.g., Donguy and Meyers, 1996; Lumpkin and Johnson, 2013) the major systems are as follows. At the equator, water flows eastward in the Equatorial Undercurrent (EUC), originating in the western equatorial Pacific (WEP) as part of low-latitude western boundary currents (Stellema et al., 2022 and references therein). The salinity of the EUC is fresher in the central Pacific than in the WEP and this has been attributed to the addition of North Pacific waters, possibly via subtropical cell transport, specifically in the upper layers of the EUC (Kuntz and Schrag, 2018). This addition has been suggested to result in the CEP reflecting a greater Northern Hemisphere influence relative to the eastern equatorial Pacific (Kuntz and Schrag, 2018). Wind-driven Ekman divergence along the equator brings EUC water to the surface, cooling surface waters and increasing their nutrient content, including micronutrients like iron sourced from the shelves around Papua New Guinea (Slemons et al., 2010). As waters upwelled from the EUC move northwards at the surface, nutrients are consumed by primary producers and productivity decreases along the flow path creating a notable, and persistent, pattern of decreasing chlorophyll concentrations along the latitudinal transect of cores examined in this study (Acker and Leptoukh, 2007; Rousseaux and Gregg, 2014 see Fig. 1). At ~2°N there is also a contribution of surface waters from the westward-flowing South Equatorial Current (SEC) (Lumpkin and Johnson, 2013; Yang et al., 2022). Between ~2.5° and 10°N surface waters also originate in the warm WEP, and they are advected zonally by the North Equatorial Countercurrent (NECC) (Donguy and Meyers, 1996; Li and Fedorov, 2022). The zonal wind stress minimum associated with the convergence of the ITCZ allows for the eastward movement of the NECC and the current's latitudinal position is associated with the seasonal migration of the ITCZ's core (Li



and Fedorov, 2022 and references therein). The positive feedback between the location and strength of the NECC and the
135 ITCZ is key for understanding coupled ocean-atmosphere changes in this region (Li and Fedorov, 2022).

Ocean-atmosphere phenomena previously investigated at the ML1208 core sites include millennial shifts of the Intertropical
Convergence Zone during deglacial transitions (Jacobel et al., 2016; Reimi et al., 2019; Reimi and Marcantonio, 2016), glacial-
interglacial and millennial scale variations in dust deposition (Jacobel et al., 2017a), glacial-interglacial changes in nutrient
utilization and export production (Costa et al., 2017, 2016), abyssal ocean respired carbon storage (Jacobel et al., 2020, 2017b),
variations in ENSO strength across the modern, Holocene and glacial period (Rongstad et al., 2019; Rustic et al., 2020; White
et al., 2018), and last glacial sea surface temperatures (Monteagudo et al., 2021).

## 3 Methods

### 3.1 Quantification of Coarse Fraction

ML1208 core samples for 06BB, 16BB, and 35BB were obtained from the Lamont-Doherty Earth Observatory Core
Repository and sent to the Facility for Oceanographic Research @ Middlebury College (FOR@M). Samples were frozen and
then freeze dried for at least 48 hours. Dry samples were subdivided to preserve an aliquot of bulk sediment for analyses and
each sample subset was weighed. Sample residuals of 8-10 g were agitated for ~20 minutes in 20 ml Nalgene wash bottles
with 15 mL of tap water to allow for disaggregation of clays. Bulk samples were wet sieved through 63 μm wire mesh sieves
until the water passing through the sieves ran clear. Sieves were dried overnight in an oven at 45°C and then re-weighed to
determine the percentage of sediment larger than 63 μm (coarse fraction percentage). Samples from ML1208-26MC and 29MC
were previously analysed for coarse fraction percentage using this same methodology but those data were unpublished as the
focus of that manuscript was on indicators of surface ocean nutrient utilization, not carbonate preservation (Costa et al., 2016).

A subset of samples were dried in a laboratory oven at 45°C (not in the freeze drier) to determine if coarse fraction values were
influenced by expansion of ice during the sample pre-freezing step (Supplementary Text S1). The experiment showed no
consistent or statistically significant offsets in coarse fraction percentage between the two methods. Therefore, both are valid
methods for removing water from samples prior to washing. All other %CF data presented in this manuscript have been
obtained through pre-freezing and freeze drying.

### 3.2 Stable Isotope Stratigraphies

Coarse fractions were dry sieved to isolate the 250-355 μm size fraction for identification of adult foraminifera. Between 10-
12 *G. ruber* (white) were picked from each sample depth for stable isotope analysis. Because of the low abundances of the
benthic foraminifera *C. wuellerstorfi*, commonly used for stable isotope stratigraphies, these specimens were reserved for B/Ca
and other analyses that can only be done on benthic foraminifera. Picked *G. ruber* samples were sent for analysis to either
Woods Hole Oceanographic Institution (WHOI) or the Lamont-Doherty Earth Observatory (LDEO). Foraminifera were





analysed for δ$^{18}$O and δ$^{13}$C on a Thermo Delta V Plus with Kiel IV individual acid bath device at both LDEO and WHOI, and some samples were analysed using a VG Prism at the National Ocean Sciences Accelerator Mass Spectrometry (NOSAMS)

Facility at WHOI. All values were calibrated to the VPDB isotope scale with NBS-19 and NBS-18. Reproducibility of the in-house standard (one standard deviation) is ±0.09‰ for δ$^{18}$O and ±0.02‰ for δ$^{13}$C (for all three measurement types). This study represents 610 new, paired stable isotope measurements. Here, we focus on the δ$^{18}$O results and interpretation, but δ$^{13}$C data have also been submitted to the NOAA NCEI site for future work.

**3.3 Radiocarbon Dates**

Seventeen new radiocarbon dates were obtained via NOSAMS, on ~2-4 mg of cleaned *G. ruber* from the 250-355 μm size fraction. Dates represent both continuous flow AMS measurements and MIni CArbon Dating System (MICADAS) measurements and the type of analysis and uncertainty for each core/depth is specified in Table 1. Radiocarbon dates were converted to calendar age using CALIB8.2 and the most recent marine calibration curve (MARINE20) (Heaton et al., 2020).

The calibrated values use the standard MARINE20 reservoir age correction (no ΔR) (Lynch-Stieglitz et al., 2015), but an uncertainty of 150 years has been added to the reservoir correction. Results are reported as median ages with +/- one and two standard deviations. To facilitate use we have also compiled all published radiocarbon dates from the Line Islands cores and recalibrated them using these parameters; these data are available in Table S1.

**3.4 Age Model Construction**

Age models were constructed via synthesis of radiocarbon dates and alignment of planktonic oxygen isotope stratigraphies to the most recent benthic oxygen isotope stack (Ahn et al., 2017) following the age model approach taken by previous studies in the CEP (e.g. Jacobel et al., 2017a). Although it might be preferable to align benthic δ$^{18}$O from the new sites to the benthic δ$^{18}$O stack, previous work has shown the abundances of benthic foraminifera (*Cibicidoides wuellerstorfi*) at these sites to be

too low for both age model development and other paleoceanographic proxies such as B/Ca for deep water carbonate ion reconstructions. As the major feature of the benthic δ$^{18}$O stack (ice volume) also impacts planktonic δ$^{18}$O records we expect that alignment will capture the major features in the record.There are some features of the planktonic δ$^{18}$O record that may not be present in the stack due to regional or local variations in temperature or surface water hydrography but we expect that these variations are small relative to the glacial-interglacial cycles that are known to be a major feature of both types of δ$^{18}$O records.

Alignment between the benthic δ$^{18}$O  stack and our new records was done using the open-source MATLAB script Bayesian Inference Gaussian Process regression and Multiproxy Alignment of Continuous Signals (BIGMACS) (Lee et al., 2023). Recent work (Middleton et al., 2024) has shown that use of an automated, probabilistic alignment algorithm like BIGMACS is preferable to manual alignment because it both eliminates the potential for user-derived biases and provides robust estimates of uncertainty for each age model control point (all input points including $^{14}$C and δ$^{18}$O constraints). Because the amplitude of

planktonic δ$^{18}$O is not expected to map one to one with the amplitude of benthic δ$^{18}$O changes, BIGMACS was allowed to identify δ$^{18}$O shift and scaling parameters during alignment, as described in the BIGMACS user manual. We also increased



the uncertainty on the stack to 0.2 ‰ to allow for a more flexible match between $\delta^{18}O$ variability and $^{14}C$-based age constraints. New age models were developed for cores 06BB and 16BB, and revised age models were developed for 15GC, 18GC, 20BB, and 28BB.


## 4 Results

### 4.1 Radiocarbon Dates

New radiocarbon dates for ML1208-06BB, 16BB, and 35BB are presented in raw and calibrated (Marine20) form in Table 1, and a complete list of the ML1208 radiocarbon measurements to date (recalibrated using Marine20) are compiled in the
supplement (Table S1). Core top dates for cores 06BB and 16BB indicate that both capture sediments deposited during the Holocene (5.07 and 3.79 ka respectively), making them potentially useful for proxy calibration work. Radiocarbon dates from 16BB suggest that it is in stratigraphic order and that sedimentation has occurred at roughly linear rates over the last ~30ka. Data from 06BB show a small apparent age reversal during the same interval that oxygen isotope data appear unusually depleted. This interval is discussed below, but otherwise the sediments at 06BB appear to be in good stratigraphic order.


| ML1208 Core | Lat (DDS) | Long (DDS) | Depth of Core (m) | Top Depth (cm) | Foraminifera | Process | Accession # | $^{14}$C Age (ka) | Age Err (ka) | Calibrated Median Age (ka) | Lower 2σ (ka) | Lower 1σ (ka) | Upper 1σ (ka) | Upper 2σ (ka) |
|---|---|---|---|---|---|---|---|---|---|---|---|---|---|---|
| 06BB | 6.41 | 161.01 | 2371 | 2 | *G. ruber* | (GCHS) Gas Carb HS | OS-172568 | 4.96 | 0.08 | 5.08 | 0.24 | 0.13 | 0.15 | 0.23 |
| 06BB | 6.41 | 161.01 | 2371 | 6 | *G. ruber* | (GCHS) Gas Carb HS | OS-172565 | 4.99 | 0.09 | 5.11 | 0.27 | 0.13 | 0.14 | 0.24 |
| 06BB | 6.41 | 161.01 | 2371 | 20 | *G. ruber* | (GCHS) Gas Carb HS | OS-172566 | 11.25 | 0.12 | 12.60 | 0.32 | 0.12 | 0.12 | 0.27 |
| 06BB | 6.41 | 161.01 | 2371 | 30 | *G. ruber* | (GCHS) Gas Carb HS | OS-172569 | 14.50 | 0.16 | 16.72 | 0.47 | 0.22 | 0.23 | 0.43 |
| 06BB | 6.41 | 161.01 | 2371 | 40 | *G. ruber* | (GCHS) Gas Carb HS | OS-172564 | 19.65 | 0.24 | 22.74 | 0.54 | 0.29 | 0.25 | 0.58 |
| 06BB | 6.41 | 161.01 | 2371 | 50 | *G. ruber* | (GCHS) Gas Carb HS | OS-172567 | 23.40 | 0.34 | 26.76 | 0.76 | 0.35 | 0.36 | 0.65 |
| 16BB | 0.48 | 156.45 | 2926 | 1 | *G. ruber* | (HY) Hydrolysis | OS-158775 | 3.79 | 0.03 | 3.56 | 0.16 | 0.09 | 0.08 | 0.16 |
| 16BB | 0.48 | 156.45 | 2926 | 16 | *G. ruber* | (HY) Hydrolysis | OS-158776 | 6.40 | 0.03 | 6.66 | 0.17 | 0.08 | 0.08 | 0.16 |
| 16BB | 0.48 | 156.45 | 2926 | 36 | *G. ruber* | (HY) Hydrolysis | OS-158777 | 12.45 | 0.06 | 13.85 | 0.23 | 0.10 | 0.12 | 0.20 |
| 16BB | 0.48 | 156.45 | 2926 | 50 | *G. ruber* | (HY) Hydrolysis | OS-165256 | 16.75 | 0.11 | 19.30 | 0.36 | 0.19 | 0.17 | 0.33 |
| 16BB | 0.48 | 156.45 | 2926 | 60 | *G. ruber* | (HY) Hydrolysis | OS-165255 | 19.85 | 0.18 | 23.90 | 0.55 | 0.14 | 0.25 | 0.39 |
| 35BB | 6.67 | 160.73 | 3777 | 0 | *G. ruber* | (GCHS) Gas Carb HS | OS-172562 | 18.65 | 0.24 | 21.64 | 0.66 | 0.30 | 0.34 | 0.59 |
| 35BB | 6.67 | 160.73 | 3777 | 2 | *G. ruber* | (GCHS) Gas Carb HS | OS-172870 | 12.75 | 0.12 | 14.31 | 0.42 | 0.26 | 0.21 | 0.50 |
| 35BB | 6.67 | 160.73 | 3777 | 20 | *G. ruber* | (GCHS) Gas Carb HS | OS-172871 | 12.05 | 0.11 | 13.38 | 0.27 | 0.14 | 0.13 | 0.27 |
| 35BB | 6.67 | 160.73 | 3777 | 25 | *G. ruber* | (GCHS) Gas Carb HS | OS-172563 | 29.60 | 0.63 | 33.11 | 1.52 | 0.75 | 0.81 | 1.30 |
| 35BB | 6.67 | 160.73 | 3777 | 48 | *G. ruber* | (GCHS) Gas Carb HS | OS-172872 | 28.60 | 0.55 | 31.98 | 1.12 | 0.73 | 0.62 | 1.36 |
| 35BB | 6.67 | 160.73 | 3777 | 60 | *G. ruber* | (GCHS) Gas Carb HS | OS-172561 | 40.70 | 2.30 | 43.35 | 3.52 | 1.65 | 1.64 | 3.90 |

**Table 1. New radiocarbon dates from the Line Islands cores. Data have been calibrated using the parameters indicated in the text.**

In contrast with the data from 06BB and 16BB, radiocarbon dates for core 35BB indicate a core top that is approximately LGM in age (21.6 ka), and ages are inverted at several depths within the core. Given these age-depth reversals, and the %CF data which both suggest dynamic sedimentation patterns throughout the core (developed further in 5.1.1), we suggest that it may be challenging to interpret time series data from this site as indicative of oceanographic/climatological changes in the region. Despite intervals of apparent remobilization throughout the core (to 400 cm) it may yet be possible to use $\delta^{18}O$ to
identify limited intervals of the core that retain stratigraphic order.





## 4.2 Oxygen Isotope Data

Oxygen isotope data measured on the planktonic foraminifera *G. ruber* are displayed with depth in Fig. 2 for cores ML1208-
06BB, 16BB, and 35BB. A first glance at the δ¹⁸O data shows strong glacial-interglacial cyclicity and suggests that 16BB may
cover as many as three full glacial-interglacial cycles. The pattern of δ¹⁸O across the three cores is suggestive of sedimentation
rates that are generally comparable, with the deglacial decrease in δ¹⁸O occurring within the top ~50 cm of all cores. Core
16BB seems to have the clearest glacial-interglacial cycles, with a larger amplitude than observed at 06BB. We refrain from
discussing the δ¹⁸O further as we prefer to consider these data in the context of their new age models (section 4.3).

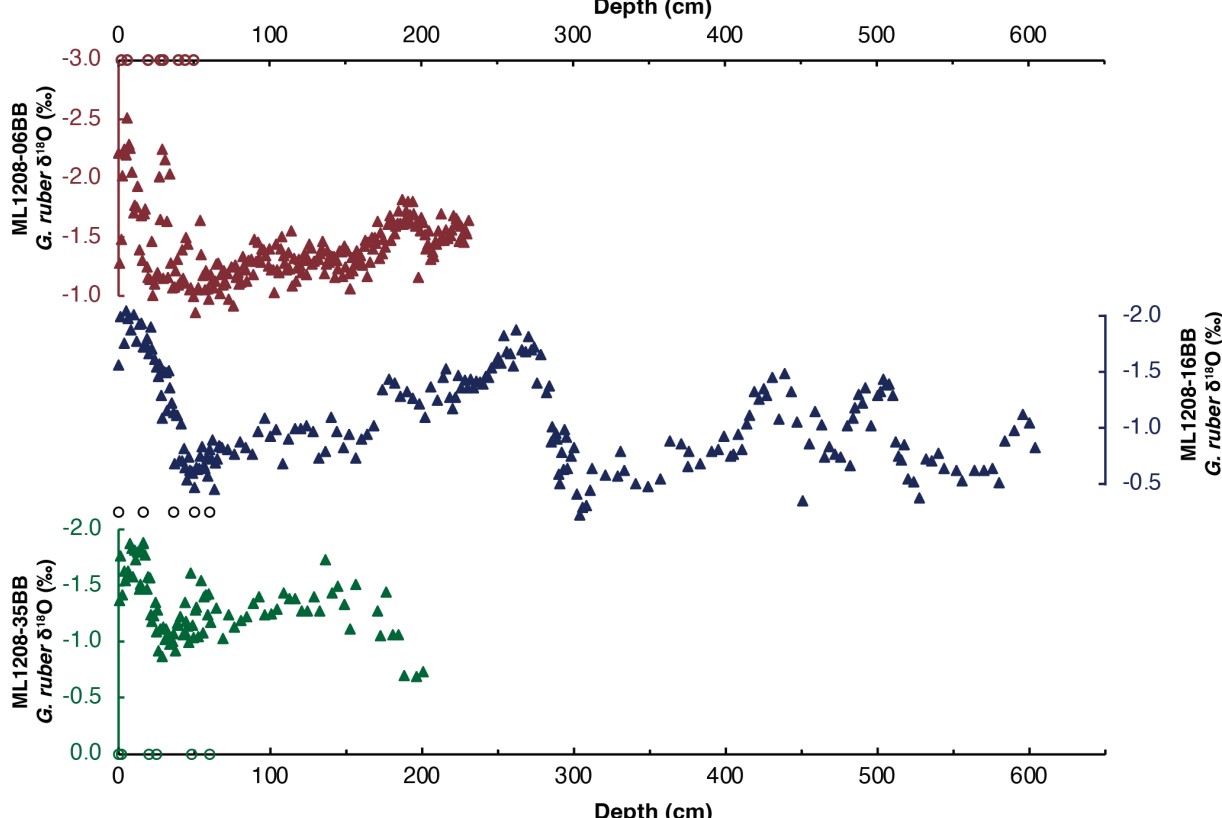

**Figure 2. δ¹⁸O data from cores ML1208-06BB (red), 16BB (blue) and 35BB (green) with depth. Triangles indicate data and open circles indicate the depth of radiocarbon age control points. All y-axes share the same scaling and are colour coded by core.**


## 4.3 Age Models and Uncertainties

Figure 3 shows the δ¹⁸O data for ML1208-06BB and 16BB on their new age models. All stable isotope and age model data
are available via the NOAA Paleoclimatology database. As previously discussed, core 35BB appears to have experienced
sufficient stratigraphic disturbance so as to make age model construction prohibitively challenging, so we do not present those
data on an age model. Data from both 06BB and 16BB show clear glacial-interglacial cyclicity with deglacial values ranging
from -0.86 to -2.24‰ for 06BB and from -0.47 to -2.02 for 16BB, with core 16BB displaying both a larger amplitude of



variability and more positive glacial δ¹⁸O values suggesting colder and or fresher conditions (discussed below). Both cores display notably positive values into the late Holocene, as previously observed in other Line Islands cores (Lynch-Stieglitz et al., 2015) and attributed to coretop remobilization and deposition, or bioturbation, of older material. In contrast with the other

Line Islands cores that have unusually positive core-top δ¹⁸O, we do not see evidence from the core-top radiocarbon dates that the samples are not late Holocene in age. Radiocarbon dates might be less sensitive to minor coretop disturbances than δ¹⁸O due to the considerably larger number of individual foraminifera measured (300-400 vs. 10-12 individuals respectively). 16BB also captures Terminations II (δ¹⁸O of -0.23 to -1.88‰) and III (-0.38 to -1.44‰). Some precessional variability is also captured in both records, especially in 16BB which shows variations on the order of 0.3‰ associated with MIS 3-4 and the MIS5

substages (Fig. 3).

Core 06BB has an interval of depleted δ¹⁸O just after the LGM, accompanied by a small radiocarbon reversal. Even though the coarse fraction data do not show obvious evidence of remobilization, the consistency of the δ¹⁸O during that interval (0.57‰ more depleted than the average of the five nearest neighbour points), combined with the radiocarbon reversal suggests

that the most plausible explanation is instantaneous deposition of this sediment as part of a slump. We have chosen to remove this short interval of deposition from the age model and to rely on the adjacent radiocarbon dates to constrain the age model (Fig. 3).

A significant advantage of the probabilistic alignment approach employed here (as described in Lee et al., 2023) is that an

alignment uncertainty estimate is provided for each data input point (¹⁴C and δ¹⁸O). The upper and lower 1σ uncertainty bounds associated with the age models for cores 06BB and 16BB are illustrated in Fig. 3, immediately below each of the δ¹⁸O time series. Time periods that are associated with maximum/minimum values (i.e., peak glacial periods, peak interglacial periods) or rapid rates of change (i.e., deglaciations) have the most straightforward 'match' with the dated target record, and thus they have the lowest uncertainties. In contrast, intervals that are not characterized by significant structure in the δ¹⁸O record tend to

have larger alignment uncertainties. The age model alignment uncertainties (1σ) for 06BB range from a minimum of ~0.4 kyr, associated with Termination I (TI) at 11 ka, to a maximum of ~3.9 kyr (high at 49 and 78 ka) with an average uncertainty of 1.9 kyr. For 16BB the minimum uncertainty is also 0.4 kyr (lowest during TI and TII) and the maximum is ~6 kyr (highest during the middle of MIS 6) with an average uncertainty of 2.2 kyr.


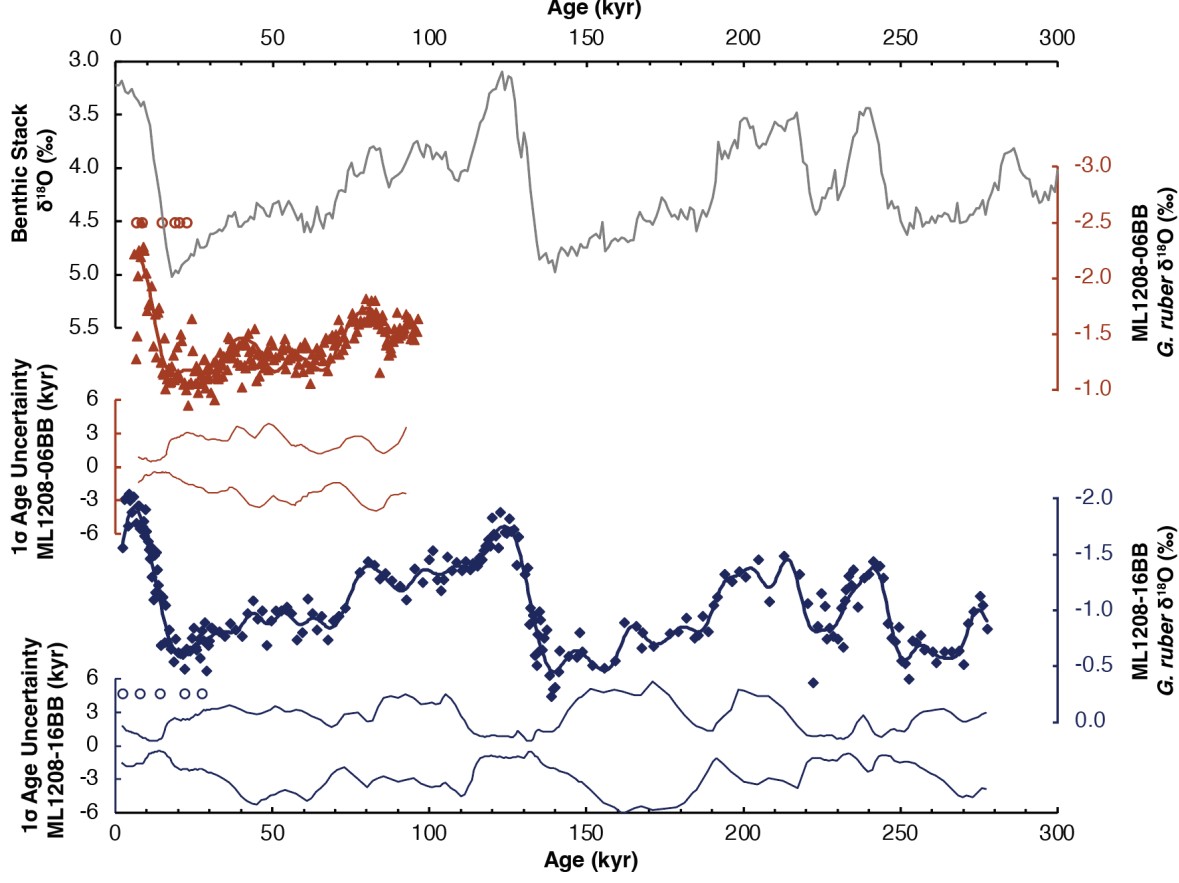

**Figure 3. δ¹⁸O data from cores ML1208-06BB (red triangles), and 16BB (blue diamonds) on their new age models. Symbols indicate data, thick lines indicate the 5 kyr lowpass filtered data, and open circles indicate the depth of radiocarbon age control points. δ¹⁸O y-axes both have the same scaling and are colour coded by core/dataset. Below each δ¹⁸O curve is the corresponding 1σ uncertainty envelope for the age model, with colours as above. The uppermost curve (grey) shows the benthic oxygen isotope stack from (Ahn et al., 2017).**

### 4.4 Coarse Fraction Percentages

The CF% for the three cores follow the relative trends expected with depth, with CF% generally decreasing with depth, marking greater dissolution and fragmentation in association with more undersaturated bottom waters (06BB and 16BB in Fig. 4 and

35BB in SF2). Cores 06BB and 16BB show broadly similar patterns through time with % CF increasing through the last glacial cycle to maxima at the LGM and then decreasing towards the Holocene. Core 06BB seems to have more pronounced variability than 16BB on multi-millennial timescales with both a greater amplitude of variability and less noise than is apparent at the deeper site. Beyond the last glacial cycle, variability at 16BB is pronounced with a sharp, well-defined increase in CF% in association with MIS6, and a minimum (much lower CF% than the Holocene) in association with the MIS7 interglacial. The

temporal correspondence between CF% and planktonic δ¹⁸O at site 16BB appears at first glance to be quite strong (SF3),



however the $R^2$ value is low (0.02), potentially due to variable lead/lag relationship between $\delta^{18}O$ and CF%. This is an important observation that may yield insight into deep ocean carbon storage and carbonate system buffering, and it should be explored with more quantitative proxies for deep ocean carbon storage and carbonate saturation state.

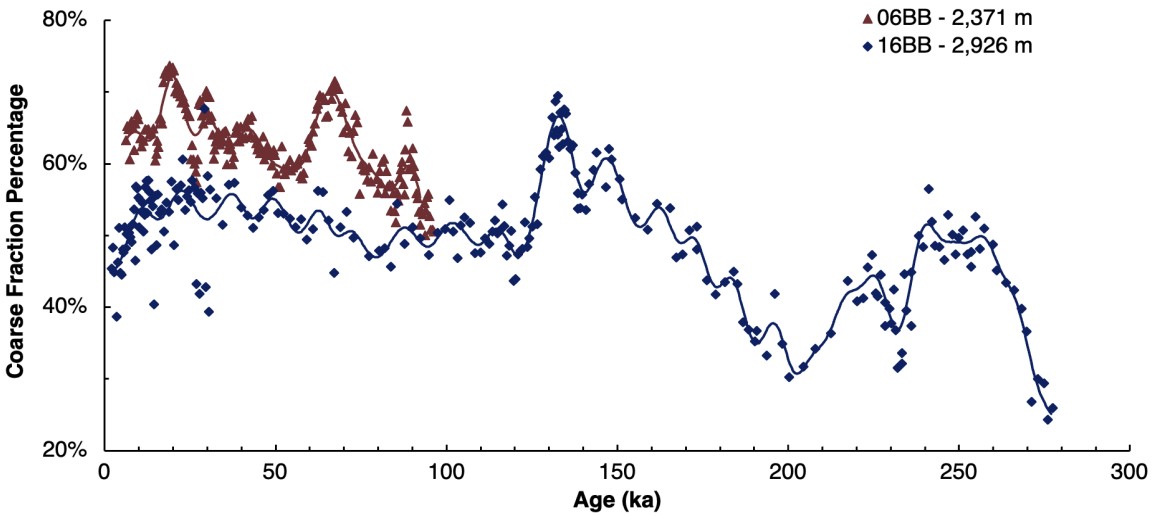

**Figure 4. Coarse fraction percentage from ML1208-06BB (red triangles) and 16BB (blue diamonds). Lines show the 5 kyr lowpass filtered data.**

The CF% for core 35BB (SF2) is consistent with our interpretation of the $^{14}C$ age reversals observed in that core. Several intervals are marked by large (20-40%) excursions in CF% on the timescale of a few thousand years (centimetres). We suggest that these rapid oscillations are unlikely to reflect the rain of material with a highly variable CF% from the sea surface, but rather reflect intervals of older sediment remobilization. This interpretation is consistent with seismic reflection survey data from the region that has been interpreted to show anastomosing channels as a result of energetic current activity (Lyle et al.,

2016). Because of the low density of whole foraminifera tests, they may be mobilized more easily than fragments, and whole tests may be redeposited in sediment waves with high %CF (Lyle et al., 2016). This hypothesis is counterintuitive relative to the classic understanding of the 'winnowing' of fines, but it appears to be well-matched with the CF% and $^{14}C$-based observations from these sites. Alternatively, the sediment remobilization may have been post-depositional and discontinuous. For example, sediment from a relatively shallower seafloor depth may have slumped onto the deep 35BB site, causing older

sediment to overlay younger sediment. No obvious unconformity was observed in the core, but it also cannot be ruled out.



## 5 Discussion

### 5.1 Age Models & Core Sedimentation Rates

The latitudinal transect of Line Islands cores has significant potential to constrain questions about paleo variability of the ITCZ, ENSO, and related features of oceanographic circulation, and so it is critical that we evaluate whether reconstructed geochemical gradients between the sites can be confidently attributed to real differences in oceanographic conditions versus the extent to which gradients may be artifacts introduced by conditions of sedimentary preservation (e.g., accumulation rate, bioturbation, and dissolution). Given the differences in nutrient delivery to these sites it might be expected that sites closer to

the equator would experience higher productivity and thus higher accumulation rates, and it might also be anticipated that shallower sites might experience better preservation during deep sea dissolution cycles. Our new $^{14}$C and $\delta^{18}$O-constrained age models, with statistically estimated uncertainties provide a strong basis for evaluating changes in sedimentation and their impacts on our climate reconstructions.

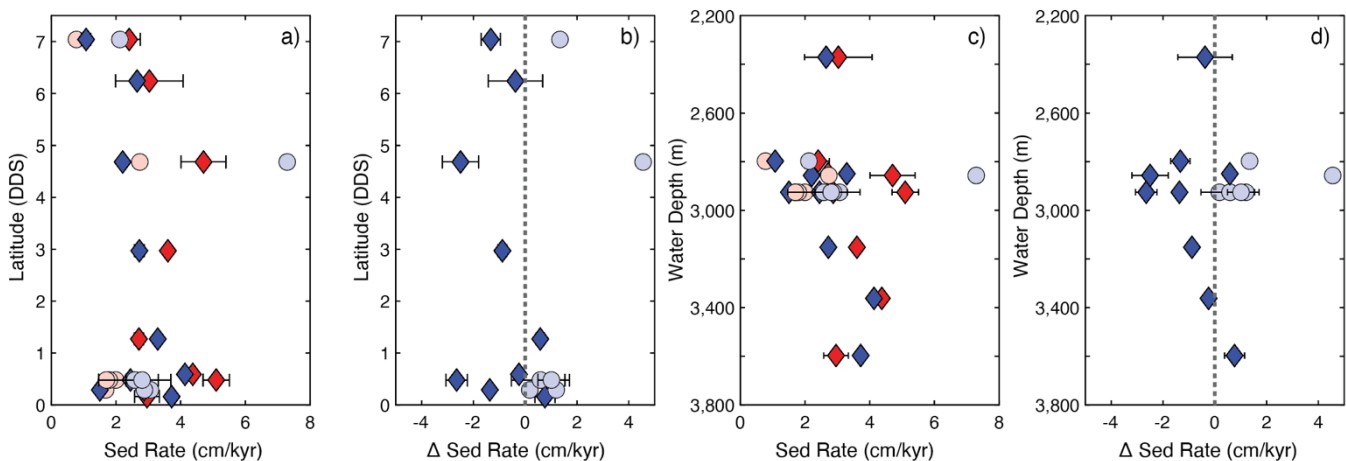

**Figure 5. Patterns of Sedimentation. a) Sedimentation rates and 1σ uncertainties with latitude for the Holocene (dark red diamonds), LGM (dark blue diamonds), older interglacial periods (light red circles), and older glacial periods (light blue circles), b) difference in sedimentation rate with latitude between the LGM-Holocene (dark blue diamonds) and other glacial-interglacial cycles (light blue circles). c) sedimentation rates with water depth (m), colours as in a, d) differences in sedimentation rate with water depth, colours as in b.**

Figure 5 shows sedimentation rates (cm/kyr) and 1σ uncertainties evaluated at Line Islands core sites over the Holocene (2-8 ka) and LGM (18-24 ka) time slices, with values and additional time slice data for MIS5 (120-126 ka), MIS 6 (138-144 ka), MIS 7 (236-242 ka), MIS 8 (248-254 ka) and MIS 9 (326 –332 ka) available in ST2. Sedimentation rates range from 0.8 cm/kyr to just under 7.3 cm/kyr with most cores showing accumulation rates in the range of 2-4 cm/kyr during the time slices chosen for comparison. Contrary to what might be expected given the gradient of equatorial upwelling (which might have been

expected to yield higher sedimentation rates closer to the equator), there is no trend in sedimentation rate with core latitude during any of the four time periods examined. This is most likely a consequence of the dynamic bottom currents acting in the



Line Islands that contribute and remove winnowed material from sites in a time-evolving way. Although this means that proxy gradients are unlikely to be systematically biased as a function of latitude, it suggests that meaningful interpretations of gradients need to consider the specifics of sedimentation at each site at each timepoint of comparison. Additionally, because

of the apparent influence of lateral sediment redistribution at the seafloor (winnowing and focusing) it is particularly important that data be flux-normalized using a constant flux proxy like $^{230}$Th or $^{3}$He (Costa et al., 2020).

### 5.1.1 Core disturbances

Core ML1208-35BB is not included in figures displaying data with depth because $^{14}$C, and %CF constraints suggest that core

has experienced episodic disturbances, and its stratigraphy is not intact. Using oxygen isotopes alone for chronologic constraints would likely not have raised suspicions of compromised stratigraphy, and likewise, intervals of episodically high %CF might be suggestive of winnowing of fines rather than true stratigraphic disturbance. However, in combination with $^{14}$C-based age constraints it appears likely that sediment of different ages is being actively mobilized and redeposited at this site, specifically unusually coarse sediments are associated with radiocarbon age reversals. Given the temporal limitations of $^{14}$C,

we suggest that pairing oxygen isotope measurements and down-core %CF measurements is the best way to comprehensively identify possible intervals of disturbance. The %CF measurements are straightforward to make during the sediment core washing process and they provide an important first order check to see if sediment redistribution may have occurred. Conversely, core 06BB clearly has intervals of disturbance that do not appear outstanding in the %CF record, but are evident in the $\delta^{18}$O data, so combining these two proxies seems like a promising strategy to identify disturbances. Core 35BB provides

an important cautionary example of the potential for significant remobilization at Line Islands cores (Lyle et al., 2016) and we choose not to interpret data from that core further.

### 5.2 Glacial-Interglacial Changes in $\delta^{18}$O and Latitudinal Gradients

Measuring planktonic $\delta^{18}$O provides both first order constraints on sediment core chronologies and interpretable information

about regional hydrography, providing additional utility over benthic $\delta^{18}$O and allowing *C. wuellerstorfi* (on which benthic $\delta^{18}$O is most often measured) to be saved for analyses that can only be carried out on epifaunal benthic foraminifera, like B/Ca to quantify variations in deep sea carbonate ion through time (e.g. Yu and Elderfield, 2007). Although all ML1208 cores show clear glacial-interglacial $\delta^{18}$O cyclicity, there are notable differences in the absolute value of $\delta^{18}$O as a function of latitude (Fig. 6). Specifically, the southern cores (cooler colours), closer to the core of equatorial upwelling, show more enriched $\delta^{18}$O

corresponding with cooler and/or more saline conditions. Given the absence of clear trends in accumulation rate as a function of latitude we are confident that these differences in $\delta^{18}$O represent real differences in sea surface conditions. Consistently





cooler/more saline equatorial conditions and warmer/fresher conditions further north are consistent with the pattern observed today and appears to have persisted for at least the last 150 kyr for which we have data at both northern and equatorial sites.

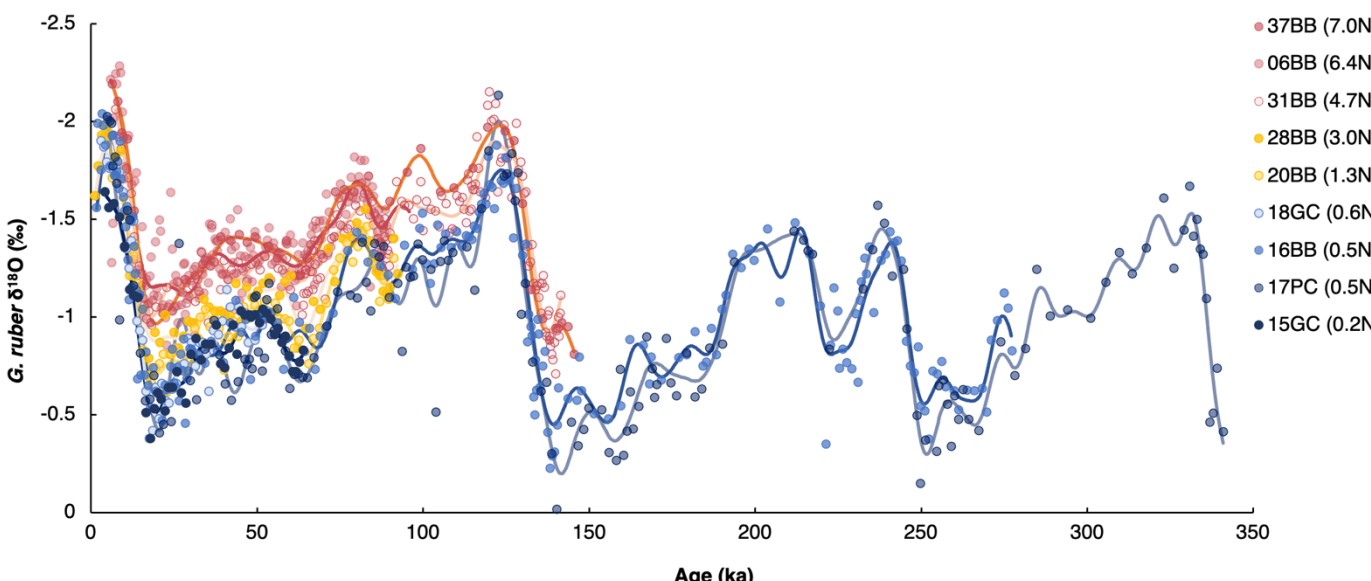

**Figure 6. ML1208 Planktonic Oxygen Isotope Records. 5 kyr bandpass filtered records of *G. ruber* δ¹⁸O from ML1208 cores 37BB, 06BB, 31BB, 28BB, 20BB, 16GC, 17PC, 16BB, and 15GC colour coded by latitude with more equatorial (further south) cores in cooler colours and more northern sites in warmer colours.**

Distinct from observations of absolute $\delta^{18}O$ values, the amplitude of glacial-interglacial variability is also inconsistent across the range of latitudes sampled, with equatorial cores appearing to show larger amplitude $\delta^{18}O$ changes during glacial periods (specifically MIS 2 and 6), enhancing the gradient of $\delta^{18}O$ with latitude during those intervals (Fig. S4). Again, because sedimentation rates at these sites do not vary as a function of latitude, it is reasonable to infer that the gradient in $\delta^{18}O$ as a function of latitude is real, and that equatorial sites did indeed experience greater cooling and/or freshening than sites further

to the north. For example, we can consider changes in sedimentation rates (Table S2), and the amplitude of $\delta^{18}O$ change (Table S3) at sites 31BB (4.68°N) and 16BB (0.48°N). Site 31BB has a higher sedimentation rate than site 16BB during both the Holocene (4.7 ± 0.7 cm/kyr vs. 2.9 ± 0.06 cm/kyr) and the LGM (2.2 ± 0.08 cm/kyr vs 1.5 ± 0.1 cm/kyr) however, the amplitude of LGM-Holocene $\delta^{18}O$ change is larger at 16BB than at 31BB (1.6‰ vs 1.1‰). For context, the $\delta^{18}O$ of seawater is thought to have shifted by +0.8‰ at the LGM due to changes in ice sheet volume (Hättig et al., 2023). This comparison also holds for

the previous glacial period (MIS6) where 16BB again has the lower sedimentation rate (3.1 ± 0.1 cm/kyr vs. 7.3 ± 0.2 cm/kyr) and shows more enriched $\delta^{18}O$ (-0.23‰ vs. -0.71‰) than at 31BB.

Despite evidence for a changing latitudinal gradient in $\delta^{18}O$, it is difficult to determine the full magnitude of increase. The quantification challenge can be seen in the $\delta^{18}O$ differences between 16BB and 17PC at MIS2 and MIS 6 (Fig. 6 and S4).

Despite being very nearly co-located, the two sites have different amplitudes of reconstructed $\delta^{18}O$ change, with 17PC



consistently showing larger amplitude variability. These differences are almost certainly due to differences in sedimentation rate, likely a function of lateral sediment addition rather than vertical flux given the proximity of the two sites. Since sedimentation rates of cm/kyr and the cm length scale of bioturbation mean the full amplitude of the signal may be underestimated by our data we consider our reconstruction of the gradient and its change to be conservative. With that in mind,

we turn our attention to evaluating the climatic driver(s) of the gradient and their variation through time.

Observation of a stronger latitudinal gradient in $\delta^{18}O$ at the LGM and MIS 6 raises the question of whether those differences can be attributed to differences in sea surface temperature (SST) and/or sea surface salinity (SSS). Previous work on a subset of the Line Islands cores suggested that the amplitude of warming the LGM to Holocene was 1.9°C across the full latitudinal

range of the Line Islands (0.22°S to 7.04°N) (Monteagudo et al., 2021). While there are demonstrable latitudinal gradients in temperature during each of those time periods, the difference between the slopes of the glacial and Holocene temperature gradients is not statistically significant (p>0.05) (Fig. S5). In other words, there is evidence that a latitudinal trend of increasing temperature from the equator to 7°N existed at the LGM, but evidence does not support a different magnitude for the gradient (change in slope) between the LGM and the Holocene. If changes in the thermal gradient cannot be invoked to explain the

$\delta^{18}O$ gradient, salinity must be primarily responsible for increasing the gradient between the LGM and Holocene. Although we do not have temperature data for MIS 6 and 5, the most parsimonious explanation is that the same mechanism(s) responsible for enhancing the latitudinal salinity gradient at the LGM were also responsible for the MIS 6 change. Based on the comparison of MIS 2-1 and MIS 6-5 provided above, the total amplitude of $\delta^{18}O$ gradient change is ~ 0.5‰. Given the modern relationship between salinity and $\delta^{18}O$ in this region (0.31‰ for a 1 unit increase in salinity) (Leech et al., 2013; Lynch-Stieglitz et al.,

2015) this suggests that salinity must have increased ~1.7 psu at southern sites relative to northern sites during these two glacial periods. Combining these observations with the temperature reconstruction from Monteagudo et al. (2021), suggests that at the LGM 31BB was fresher than the Holocene by 0.26 psu while 16BB was more saline by 1.3 psu (Fig. 7c). The sense of these changes is robust regardless of whether absolute values, mean time slice values or bandpass filtered values are used to calculate the gradient.


In considering the potential causes of salinity variations at our core sites, and inferring large-scale climatic drivers, it is important to remember that the salinity signal has been measured on the calcite of *G. ruber*, a planktonic foraminifer that calcifies in the mixed layer. Thus, salinity changes may reflect the composition of water advected to the site and/or precipitation-evaporation changes at the surface. As discussed in the introduction and background to this work, two of the key

climate phenomena that influence the surface hydrography of the CEP are changes in the mean state of the El Niño Southern Oscillation (ENSO) and the position of the Intertropical Convergence Zone (ITCZ). Here we consider the hydrographic





fingerprint of each phenomenon and show that the observed changes can be interpreted as uniquely indicative of changes in the ITCZ mean position.

Much of the water delivered to the surface of the CEP is derived from the EUC, sourced from the Western Pacific Warm Pool, and it might be reasonable 405 to infer that the CEP salinity signal was advected to the Line Islands, possibly due to changes in ENSO mean state. Previous work has shown that surface 410 water conditions were fresher in the Western Pacific Warm Pool during the LGM (Hollstein et al., 2018; Lea, 2000; Rosenthal et al., 2003), 415 possibly as a consequence of changes in regional monsoon intensity driven by high latitude ice sheet forcing. Results from proxy reconstructions and

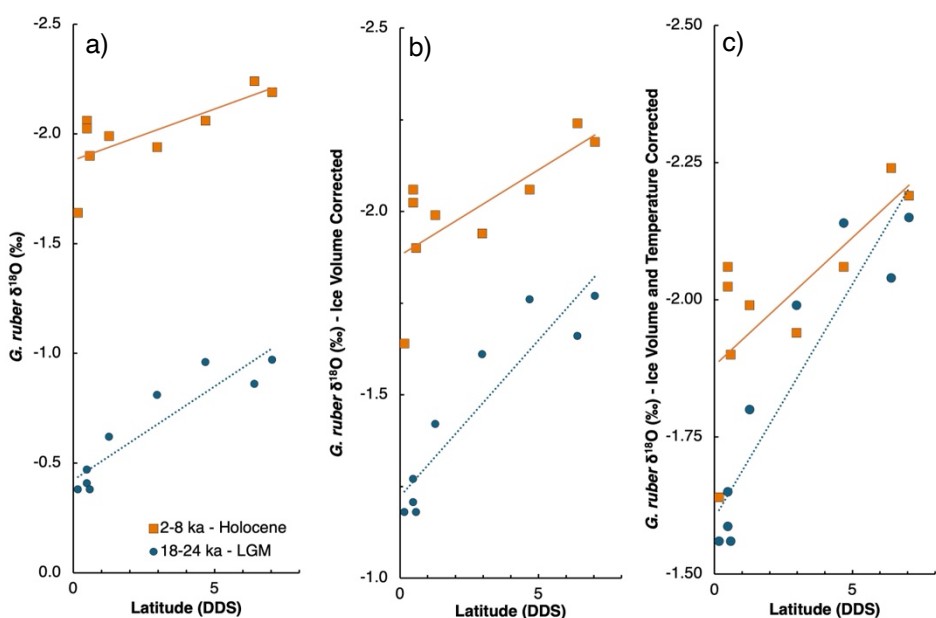

**Figure 7.** *G. ruber* δ¹⁸O gradient with latitude a) lines of best fit for the gradients over the Holocene (orange squares and solid orange line) and LGM time slices (blue dots and dotted blue line) b) the gradient corrected for a global mean seawater (ice volume) change in δ¹⁸O of 0.8‰ (Hättig *et al*., 2023) c) gradient corrected for both global mean seawater change, and the 1.9°C temperature change found by Monteagudo et al., 2021. Note that the Holocene data are unchanging in the three subplots but the y-axis range narrows from -2.5 to 0‰ to -2.5 to -1.5‰.

numerical simulations spanning the equatorial Pacific have suggested that ENSO variance was reduced during the LGM (Ford et al., 2012; Leduc et al., 2009; Thirumalai et al., 2024), and that the mean state was more El Niño-like with a deep EEP thermocline and a reduced zonal SST gradient across the Pacific (Ford et al., 2018) although not all reconstructions agree (e.g. Koutavas and Joanides, 2012). Thus, a more El Niño like mean state at the LGM (weaker zonal SST gradients, weaker winds), coupled with overall freshening of source waters in the Western Pacific Warm Pool, might be expected to manifest as a 425 freshening of equatorial waters. This pattern is the opposite of what we have reconstructed for equatorial site 16BB at the LGM, and inferred for MIS 6, suggesting that advected changes, and changes in ENSO mean state, while undoubtedly important for the broader equatorial Pacific, are not primarily responsible for changes in the gradient of surface hydrography on glacial-interglacial timescales at our Line Islands sites. Given that the signal we observe varies as a function of latitude it





might also be more reasonable to infer that a climate phenomenon with north-south variability is responsible, rather than a

phenomenon like ENSO with a primarily zonal imprint.

The key feature of our reconstructed hydrographic pattern at the LGM, and that inferred at MIS 6, is a relative increase in salinity at the equator, and freshening at ~5°N. A southward shift of the ITCZ relative to its modern-day position at 7°N in the Line Islands has both local and basin-wide implications that can comprehensively explain these reconstructed changes in

salinity. First, a freshening at our northernmost sites is consistent with a southward shift of the mean annual position of the ITCZ precipitation centroid. Equatorial sites, like 16BB, would not be expected to have experienced freshening because they are positioned too far south and would not be directly impacted by the ITCZ shift. Today, lower salinities at equatorial sites in the CEP, relative to the WEP (Kuntz and Schrag, 2018), are attributed to the contribution of low-salinity North Pacific waters which are added to the EUC (Nie et al., 2019) via shallow, subtropical cell transport (Perez and Kessler, 2009). A reduction in

the volumetric contribution of North Pacific-sourced waters to the EUC, or more saline conditions in the North Pacific would be expected to increase the salinity of conditions at equatorial sites in the CEP, without a comparable signal in the WEP. This hypothesis is consistent with previous work showing that the North Pacific waters may have been more saline at the LGM (Rae et al., 2020) possibly due to increased Ekman suction within the South Pacific Gyre, and a southward shift of the jet stream (Gray et al., 2020 and references therein), hypotheses internally consistent with our inference of a southward shifted

ITCZ. Thus, freshening at northern Line Islands sites and increasing salinities at equatorial sites are both consistent with a southward shift of the ITCZ.

Previous work on the Line Islands and other CEP cores has attempted to constrain the last deglacial and penultimate deglacial movement of the ITCZ by quantifying dust fluxes (Jacobel et al., 2017a, 2016) and the isotopic composition of dust (Kim et

al., 2024; Reimi et al., 2019; Reimi and Marcantonio, 2016). Results for the LGM suggest that the mean annual position of the ITCZ was shifted southwards relative to its mean annual position today (~7°N) by about ~3° (Kim et al., 2024), but that it did not reach the equator (Reimi and Marcantonio, 2016). These results are entirely consistent with our new, salinity-based inferences of ITCZ movement.

**6 Conclusions**

Given the significance of the equatorial Pacific as a driver of the global climate system, it is a critical region for paleoceanographic reconstructions. Specifically, the central equatorial Pacific is an understudied region with significant potential to help us understand coupled ocean-atmosphere climate phenomena such as ENSO and the ITCZ. Key to pushing forward this research are sediment cores with well constrained age models that permit millennial-scale and higher

reconstructions of climate and oceanographic history and that can help determine the mechanisms and dynamics of change. We have presented two new, statistically constrained age models for core sites in the central equatorial Pacific that extend back through two glacial cycles, providing new opportunities for paleoclimate study. Our results show the importance of

considering multiple chronologic controls ($\delta^{18}$O and $^{14}$C) and sedimentological properties (%CF) in evaluating stratigraphic integrity and assigning age-depth relationships. Specifically, our results suggest that %CF can be an important indicator of

stratigraphic disturbance, especially at sites where episodic current activity redistributes sediments of similar age downslope.

In addition to the new chronologic constraints provided by this work, we have also updated and recalibrated the database of radiocarbon dates from previously studied Line Islands cores. In collating new and existing data, and creating new age models for key cores, we also synthesized planktonic oxygen isotope data to show that the glacial salinity gradient between the equator

and 5°N was strengthened during at least the last two glacial cycles. By considering contemporaneous records from the western equatorial Pacific, we have identified a prolonged southward shift of the ITCZ during the LGM, and likely also during MIS 6, as responsible for the strengthened glacial salinity gradient.

Several of the ML1208 Line Islands sites that were initially studied (Lynch-Stieglitz et al., 2015) have already been depleted

in key intervals, but many more unstudied questions remain. Here we provide high-resolution chronologies and recommendations for future work that will expand the material available for analysis and help to resolve key questions about the central equatorial Pacific and its role in the global climate system.

**7 Data Availability Statement**

All authors contributed to data collection and the final manuscript. Work was conceived by A.W.J. and K.M.C. Data were analysed by A.W.J, and K.M.C. The manuscript was written by A.W.J. with contributions from all coauthors. All data sets can be found on the National Oceanic and Atmospheric Administration (NOAA), National Centers for Environmental Information (NCEI) Paleoclimatology Database: https://www.ncei.noaa.gov/access/paleo-search/study/39419.

**8 Acknowledgments**

This work was funded by NSF OCE-210300 to A.W.J., NSF OCE-2103031 to K.M.C., and startup funding to A.W.J. from Middlebury College. NOSAMS acknowledges OCE-1755125. The authors thank the chief scientists, crew, and scientific party of the R/V Marcus G. Langseth for their work recovering these sediments, Nichole Anest and the curatorial team at the Lamont-Doherty Earth Observatory Core Repository for sampling assistance, and undergraduate members of the Facility for

Oceanographic Research @ Middlebury (FOR@M) for sample preparation.

**9 Conflict of Interest**

The authors declare no conflicts of interest relevant to this study.




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
