# Peer review of "New Controls on Sedimentation and Climate in the Central Equatorial Pacific Ocean"

_EGUsphere, 2024_

## Author Comment (AC1)

This study by Jacobel et al. provides age modeling and stratigraphic information about a series of Central Equatorial Pacific sediment cores. Using new radiocarbon (14C) ages and over 700 new D18O values, dated stratigraphies for three cores, new data for two cores, and context for an additional seven from the Line Islands are presented. The authors use a previously-published modeling tool (BIGMACS) to integrate and align their findings and provide statistical rigor. The study proposes using the coarse fraction (CF%) to better characterize the stratigraphies and to identify possible discontinuities, slumps, or other age inconsistencies in these cores to further improve their usefulness as paleoclimate proxies. Finally, the authors use these findings to identify potential changes in the position of the ITCZ over the last glacial/interglacial cycle based on latitudinal changes in D18O related to precipitation.

The manuscript is well-written and descriptive. It provides excellent background oceanographic and depositional background information for the central equatorial Pacific region around the Line Islands, and it will be a useful addition to the literature. The approach is straightforward and the interpretations follow from the data presented. As one who is familiar with the Line Islands region and one who has used these sediments a great deal, I found myself nodding in agreement with the authors on many points, and appreciate the work here to detail the age models for these cores and demonstrate the utility of such models for paleoclimatic interpretation.

> We thank the reviewer for their positive assessment of the manuscript and its utility for other researchers working in the Line Islands region.

Some detailed items below:

Figure 1 (caption) - the authors reference 72PC, but I didn't see that labeled on the map. The wording around this is a bit confusing - I believe the authors are saying that is the only other core, in the map but outside of the Line Islands, that has multiple glacial-interglacial cycles? Perhaps that could be clarified.

> Yes, the reviewer interpreted this correctly. 72PC is one of the orange sites indicated in the inset map (not in the Line Islands region). We can clarify the caption and main text to more clearly explain the location of 72PC and why it is explicitly mentioned, but the reviewer is indeed correct that it is one of the few cores from the Central Equatorial Pacific that goes back more than on glacial-interglacial cycle.

Age Modeling - Why the additional deltaR uncertainty ? Using the Calib marine reservoir age viewer, there are several "nearby" locations (within 1000km, which, given the spatial extent of the cores, is not overly broad) that provide an average deltaR of ~130 years (but a smaller uncertainty, and are corrected for use with the Marine20 calibration. While this likely doesn't change the results that much, it would likely provide more precision in the earlier portions of the age models and prevent mis-alignment with cores where such reservoir corrections are used.

> We agree with the reviewer that reservoir correction data are available in the region, but we are somewhat more skeptical as to the consistency of the DeltaR data. Including the

points from Hawaii the DeltaR ranges from -184 to 136 years. We therefore adopted the standard DeltaR used for Marine20 (which is to say a DeltaR of 0), which is also consistent with the Line Islands stratigraphy paper published by Lynch-Stieglitz et al., 2015. Given the large range of DeltaR values we felt it was appropriate to use a somewhat larger DeltaRError of 150 years. We felt that incorporating a larger uncertainty was appropriate given that we wanted our sedimentation rates to be robust but include a reasonable error window.

We agree with the reviewer that using a different DeltaR (for example the average value for the Line Islands) would not change the age models that much, and we prefer the larger uncertainty to account for the large range of DeltaR values measured in the region.

(near line 220) - The optimism around core 35BB may be unwarranted. Given the uncertainties between CF%, D18O, and age, and with the observed inversions and sediment mobilization, it would seem that 35BB would require significant effort and meticulous dating to be useful. Opting to not use this for interpretation is prudent.

We agree with the reviewer that making sense of 35BB would likely be challenging (as indicated in the text), but we leave open the possibility of pursuing certain intervals further because the core is one of the deepest ML1208 cores and may provide limited, but important insights into carbonate preservation and deep ocean carbon storage. We can add text to clarify why we think it might be worth doing some additional work on 35BB to fully explore its potential utility before discarding it.

Line 234 - The line "We refrain from..." Is a bit odd there, since the following section (immediately after that sentence) is the section referenced for more information.

We agree with the reviewer and can remove or reword this sentence.

251 - "small radiocarbon reversal" - the magnitude and core depths should be quantified and probably further explained here. The table (Table 1) shows 6 radiocarbon ages in stratigraphic order - is the reversal due to a previous 14C age not in this table? Given this, and the spread of D18O data from 50-0cm (Figure 2), it is important to clarify and quantify this (these?) reversal(s), especially if this core will be used for future LGM-Holocene work. Some additional explanation here is likely in order to detail what section was excised and using what parameters (and do any other sections fit those parameters).

The reviewer is correct that the new [14]C data generated on 06BB are all internally consistent which is why the 'reversal' does not appear in Table 1. The dates that reflect the possible reversal are included in supplementary Table S1 (ML1208-06BB 28-29cm and 30-31cm) and the former comes from the Monteagudo et al., 2021 study. We can more explicitly direct the reader to these data in the text. We can also add detail to clarify the magnitude of the reversal (considering the 2 sigma uncertainty on the points at 28 and 30 cm the offset is only ~0.45 ka). Because of uncertainty over which of these dates was more reliable, because of noisy d18O data suggestive of a slump, and because of apparently reliable dates on either side of this interval (consistent with a linear

sedimentation rate) we excluded both points from our BIGMACS age model. We can add these details to the text and update Figure 3 to show only the [14]C dates used for the age model construction.

272 -CF% of 16BB is said to generally be decreasing depth. However, the highest CF% is @ 150ka (300cm?), and the distal end of the core (near 250ka, 500cm) is the same (~50%) as the upper 20ka.

We can clarify the first sentence of 4.4 to indicate that we meant the %CF is lower with greater water depth. Generally, the %CF decreases with increasing depth in the cores, but as the reviewer points out the G-IG oscillations are superimposed on this trend. We can clarify this description.

340 - the argument for saving wuellerstorfi for B/Ca is made multiple times (see 165, 190).

Agreed, we can mention this just once on line 165. We wanted to thoroughly justify our use of *G. ruber* but were perhaps excessive.

345 (but also other times) - is the relationship between salinity and D18Osw defined in this paper? Several times the argument is made that enriched D18O is indicative of cooler and/or more saline conditions, but a line or reference that details that relationship would strengthen this claim. Note: this is described later on line 384. Perhaps this relationship should precede the interpretations.

We will move this detail and associated references up higher in the paper and make clear that we are relying on previous measurements from the region to underpin these statements.

394 (and others, like 426)- here you discuss the "mean state of the El Niño Southern Oscillation" or "ENSO mean state"- but then proceed to discuss the mean conditions of the tropical Pacific (e.g., "weaker zonal SST gradients, weaker winds"). If this is not about ENSO variability (e.g., Thirumalai '24 or Ford '15), where the point is whether ENSO is more or less *active* (e.g., variability), then would it be more accurate for these conditions to be called the mean state of the tropical Pacific? And while some have hypothesized and/or described a link between tropical Pacific mean state and ENSO variability, being specific here on which is meant would be more precise.

We agree with the reviewer and thank them for catching this important distinction. We do indeed mean to reference the mean state of the tropical Pacific. Given the resolution of the data presented here we are not able to speak to ENSO variability.

431+ - In my opinion, the interpretation as ITCZ fits the data best, and is consistent with both the data presented and previous work. These two paragraphs describe this relationship and do a convincing job of linking the data with this interpretation. I look forward to sharing this with my students once this is published.

We sincerely appreciate that the reviewer found our analysis to be compelling. We spent quite a bit of time working to ensure that our interpretations were clear and dynamically-driven so we are delighted that the reviewer found the text to be convincing and clear.

Overall, the authors present a convincing case that ITCZ migration can explain the D18O gradients observed in their cores. The use of a model to integrated 14C and D18O into age models provides uncertainty bounds for the age models. While parts of core 16BB appear to have some stratigraphic ambiguity, these do not appear to have significant implications for the interpretation of ITCZ migration when comparing Holocene and LGM /MIS 6 sections. The minor issues noted are meant to enhance clarity and exposition. Of these, the most important in the scope of this work is detailing the reversals of 16BB, which I believe would strengthen the results.

We appreciate the reviewer's close reading of the manuscript and their suggestions to improve the clarity of the writing and interpretation. We believe we can readily address the reviewer's thoughtful comments, and we look forward to making these data and interpretations available to the broader community. Thank you!

---

## Author Comment (AC2)

Response to Reviewer 2

The manuscript by Jacobel and co-workers provides a well-presented and well-justified study on sediment cores from the Line Island region. Using new and available stratigraphic data they provide new age model constraints for the studied cores and come up with an interpretation on how the underlying stable oxygen isotope trends along a latitudinal transect are influenced by ITCZ movement.

Overall, this is a well executed study that is easy to read and follow and it provides a nice example that age model construction is the backbone of subsequent paleoclimatic and paleoceanographic reconstructions. Based on the current version, I recommendpublication of the study after some minor changes and corrections.

> We sincerely appreciate the reviewer's positive assessment of the manuscript and thank them for their careful reading and suggestions.

My main comments and suggestions are (in the way they appear in the text):

1) Figure 1: for someone not familiar with the Line Island region, it would be helpful if the authors could change the inset map to one that shows a broader picture that locates the islands in the central Pacific. Furthermore, core 72PC is mentioned in the caption but it is not shown in the map. Also, the caption says "of the core sites pictured,..." while I do not see any core pictured in the inset.

> In creating the figure and the inset we struggled to provide both the appropriate level of geographic context (inset) and resolution (basemap) so that the reader could see the larger context and the spatial distribution of the Line Islands cores. We settled on including the Hawaiian Islands in the inset because they are one of the few recognizable features in the central Pacific. We can modify the caption to explicitly identify this point of reference.

> The small orange dots in the inset are the cores to which we refer in the text and caption. We struggled to make the dots sufficiently large as to be easily visible, while still allowing for illustration of their spatial distribution. We can experiment further with the size and color of the dots to try and make them clearer.

> As mentioned in our response to Reviewer 1, we will plan to clarify our reference to 72PC.

2) Line 109: also here the cores in the inset are mentioned.

> We will work to enhance the visibility/contrast of these cores.

3) Chapter 4.1: Here you mention a reversal in the 14C ages. But in Table 1 there is no reversal visible in the data. Guess these are in the supplemental information, but since this has a prominent meaning in the main text, I would suggest to add the old 14C dates also in Table 1 and lable them accordingly.

Please see the response to Reviewer 1 for our detailed response to this point. We agree that the reversal and our handling of the data are worthy of further clarification and an additional reference to SI Table 1, but we prefer not to present other author's $^{14}C$ dates in Table 1 as it would become quite unwieldy. We can clarify in ST1 which dates were used in the age model construction.

4) Lines 252-257: Is there sedimentological evidence that is supporting the deposition of a slump in this part of the core?

We have examined the MST data and core logs for 06BB (Data from NOAA: MST, logs), but given the homogeneity of the foram oozes that characterize the Line Islands we suggest that it is challenging to find sedimentological evidence of a slump, especially on this scale (9 cm). We argue that the consistency of the $^{14}C$ reversal and noisy $\delta^{18}O$ over the 9 cm interval are strongly suggestive of a small slump.

5) Chapter 5.1.1: Also here I would recommend to add sedimentological information about the potential slumps and the reworking if there is any observation/data available that could support the atatements in this chapter.

Core logs are available (from NOAA here) and suggest some graded sand intervals in 35BB consistent with the larger slumps observed for that core. The MST data do not appear to add interpretive power, likely because of the sedimentological consistency of the foram oozes. We can add this information to the MS.

6) Lines 350-370: Here the authors repeat a coupe of times that there is a connection of some sort between the observed amplitude in their data and the sedimentation rates. Furthermore, in lines 366 to 367 they argue exactly the opposite compared to the text before. For me, there is no clear connection between these two things and I would like to ask the authors to revise the text to make the reasoning for their interpretation mor eclear to the reader. Maybe it is something ´very siomple, but I do not get it from the text provided.

We believe we can help clarify by adding text (starting on line 347) to more clearly articulate our logic. Changes in the amplitude of a recorded signal *might* be expected to occur as a function of sedimentation rate, with lower accumulation rate sites experiencing greater bioturbation or aliasing due to sampling that might tend to lower the amplitude of the signal. Here, we are assuring the reader that we can eliminate that explanation for variations in signal amplitude with latitude, because our results show that there is no trend in accumulation rate as a function of latitude. We further illustrate that point by doing an intercomparison of two key cores. Our analysis suggests that the differences in the absolute values AND amplitudes of the signals measured at different latitudes are a function of real differences in environmental conditions, rather than preservation.

7) As a non-native speaker, there seems to be something missing in the sentence of line 374 "...the amplitude of warming the LGM to Holocene..."

Yes, this sentence should read "Previous work on a subset of the Line Islands cores suggested that the amplitude of warming from the LGM to Holocene was 1.9˚C across the full latitudinal range of the Line Islands (0.22°S to 7.04°N) (Monteagudo et al., 2021)."